# Predictors of obesity among school-age children in Debre Berhan City, Ethiopia

**Abebe Nigussie Ayele** [1] *, **Alemayehu Gonie Mekonen**[1], **AbdulWahhab Seid**[1], **Esubalew Guday Mitikie** [2], **Abrham Demis Abayneh**[3], **Mitiku Tefera Haile** [3]

1 Department of Nursing, Debre Berhan University, Asrate Woldeyes Health Science Campus, Debre Berhan, Ethiopia, 2 Department of Nursing, Debre Berhan Health Science College, Debre Berhan, Ethiopia, 3 Department of Midwifery, Debre Berhan Health Science College, Debre Berhan, Ethiopia

* abebe2014nigussie@gmail.com

## Abstract

### Background

Obesity causes a serious diet-related chronic disease, including type-2 diabetes, cardiovascular disease, hypertension, osteoarthritis, and certain forms of cancer. In Sub- Saharan Africa including Ethiopia, most nutritional interventions mainly focused on a child undernutrition and ignored the impacts of obesity among children. In Ethiopia, the magnitude and associated factors of obesity among school-age children were not clearly described. Therefore this study assesses the predictors of obesity among school- age children in Debre Berhan City, Ethiopia, 2022.

### Methods

A cross-sectional study design was conducted from June to July, 2022. Participants were selected by using multistage sampling method. Data were collected using pre-tested and structured questions. Data were coded and entered in Epi-data version 4.6 and exported and analyzed using SPSS version 25.

### Result

A total of 600 children were participating in the study. The prevalence of obesity was 10.7% (95% CI: 8.3, 13.2). In this study, attending at private school (AOR = 4.24, 95% CI: 1.58, 11.32), children aged between 10-12years (AOR = 2.67, 95% CI: 1.30, 5.48), soft drink available in home (AOR = 2.27, 95% CI: 1.25,18.13), Loneliness (AOR = 1.67 95% CI: 1.12, 3.15) and mothers with occupational status of daily labour (AOR = 8.54 95% CI: 1.12, 65.39) were significantly associated with childhood obesity.

### Conclusion

In this study, the overall magnitude of childhood obesity was (10.7%) which means one in eleven children and relatively high as compare to the EDHS survey. Therefore, more attention should be given to strengthening physical activities, providing nutritional education, and creating community awareness about healthy diets as well as other preventive measures.

**Data Availability Statement:** All relevant data are within the paper and its Supporting Information files.

**Funding:** The authors received no specific funding for this work.

**Competing interests:** The authors declare that they have no competing interests exist.

## Introduction

Obesity is defined as abnormal or excessive fat accumulation that may impair health and well-being of people [1]. Body mass index (BMI) is a simple index of weight-for-height that is commonly used to classify overweight and obesity in adults. In children, BMI for age greater than or equal to 95th percentile is obesity [2]. Childhood obesity are rising because of increased consumption of sweet food, physical inactivity, urbanization and change in transportation [3]. Obesity is more prevalent in urban children [4].Childhood Obesity (CHO) is increasing at an alarming rate in both developed and developing countries [5]. In Bangladesh, a higher prevalence of obesity was observed among boys(67.1%) than girls (35.7%) [6]. Similarly in Qatar, children (17.4%) were obese [7]. Obesity in children are a major health problems in low-income countries like Ethiopia [8]. Obesity causes a serious diet-related chronic diseases, including type 2 diabetes, cardiovascular disease, hypertension, osteoarthritis, and certain forms of cancer [9]. Childhood obesity is distressing problems because of their potential long-lasting consequences in adulthood [10]. Obese children were more in psychosocial distress than healthy weight children, and more prominent in girls than boys [11]. Obesity is a global public health issues in both developed and developing nations at different speed and pattern [5]. Obesity cause more deaths than underweight in the world [12]. Obesity now rank as the fifth leading cause of global mortality, and one of the uppermost health challenges and risk factors for chronic diseases in the 21$^{st}$ century [13]. World Health Organization in 2016, over 2 billion people suffer from overweight and obesity, of which 340 million are children [5]. World Obesity Federation (WOF) estimates that about 254 million children aged in 5–19 years old will be obese by 2030 in the world [14]. It is not only a high-income countries problem, the rate of increase in childhood overweight and obesity is 30% higher in low and middle-income countries [15]. The costs of obesity and obesity-related disease are increasing. It is estimated that the total cost of overweight &obesity to health services globally is US$ 990 billion per year, over 13% of all healthcare expenditure [16]. Another impact related to obesity is poor school performance of students in their academic scores[10]. Childhood obesity is predictors for psychological problems like anxiety or depression, social problems like stigma, and poor quality of life [17]. Obesity is associated with an increased risk of morbidity and mortality as well as reduced life expectancy by 5–20 years [18]. In Africa, the number of children who were obese had more than doubled from 5.4 million in 1990 to 12.6 million in 2015 [17]. In Sub-Sahara Africa (SSA), about 13.6% of school -age children were obese [19]. In Ethiopia, the magnitude of obesity are increasing from time to time because of urbanization, changing of eating habit and mode of transportation, sedentary life, and physical inactivity [3]. For example in Addis Ababa, the prevalence of overweight and obesity were 39.1% [20]. In Sub- Saharan Africa including Ethiopia, most nutritional interventions mainly focused a child under nutrition and ignored the impacts of obesity among children [21]. In Ethiopia, the magnitude and associated factors of obesity among school-age children were not clearly described. In my experience, I see a lot of obese people in Debre Berhan city. However, no study was found during the literature review period that had been shown Debre Berhan city. Therefore, this study assesses the magnitude and associated factors of obesity among school-age children in Debre Berhan city.

## Methods

### Study design, setting, and period

Cross-sectional study was conducted in Debre Berhan City from June to July 30, 2022 G.C. Debre Berhan City is located 130-kilo meters from Addis Ababa, the capital city of Ethiopia

and 696 km away from the Bahirdar city Regional state of Amhara. The total population of the city is 310,254 (51). According to Debre Berhan city Administration Educational Bureau, there are totally of 30 primary schools (1–8 grade), of which 16 are governmental while the rest 14 are private schools. There are 13,977 School age children in the city. Among these, 7049 are females and 6928 are males.

## Populations

**Source population.** All school-age children lived in Debre Berhan city and enrolled in from grade 1-6th in the 2021/2022 academic year.

**Study population.** All students from selected primary schools in Debre Berhan city in the academic year, 2021/2022.

## Inclusion and exclusion criteria

**Inclusion criteria.** All students from grades 1-6th who permanently living in the city and enrolled in school in the academic year, 2021/2022

**Exclusion criteria.** A student with evidence of deformity in limbs or spine, and edematous condition were excluded from the study.

## Sample size determination and sampling techniques

The sample size was determined using double population proportion formula.The sample size is calculated by using Epi info version 7 statistical package by considering vehicle available, school type, having friends, sweet food preference, eating breakfast regularly, watching TV/ movies, duration of watching TV/movies as the major predictor variables.

**Where.**

**P1**: is proportion of exposed with the outcome

**P2**: is proportion of non-exposed with the outcome;

**Z $\alpha$/2**: is taking CI 95%;

**Z$\beta$**: 80% power and, r is the ratio of exposed to non-exposed 1:1

So that the final sample size was 612.Stratified-multistage sampling technique was used to select the study participants. Schools and students constituted the primary sampling unit (PSU) and secondary sampling unit (SSU) respectively. The primary sampling unit (PSU) was selected after stratifying schools into governmental and private schools. The secondary sampling unit (SSU) was selected students for each selected school by considering the identification number (ID) of students as the sampling frame in the roster by using systematic random sampling methods with ' K' intervals of nine (k = 9) and the parents of the selected students were included.

Variables (*Show Fig 1*).

**Dependent variable.** Childhood obesity.

## Independent variable

- Socio demographic factors

- Behavioral factors

- Psychological factors

- Parenting style and characteristics

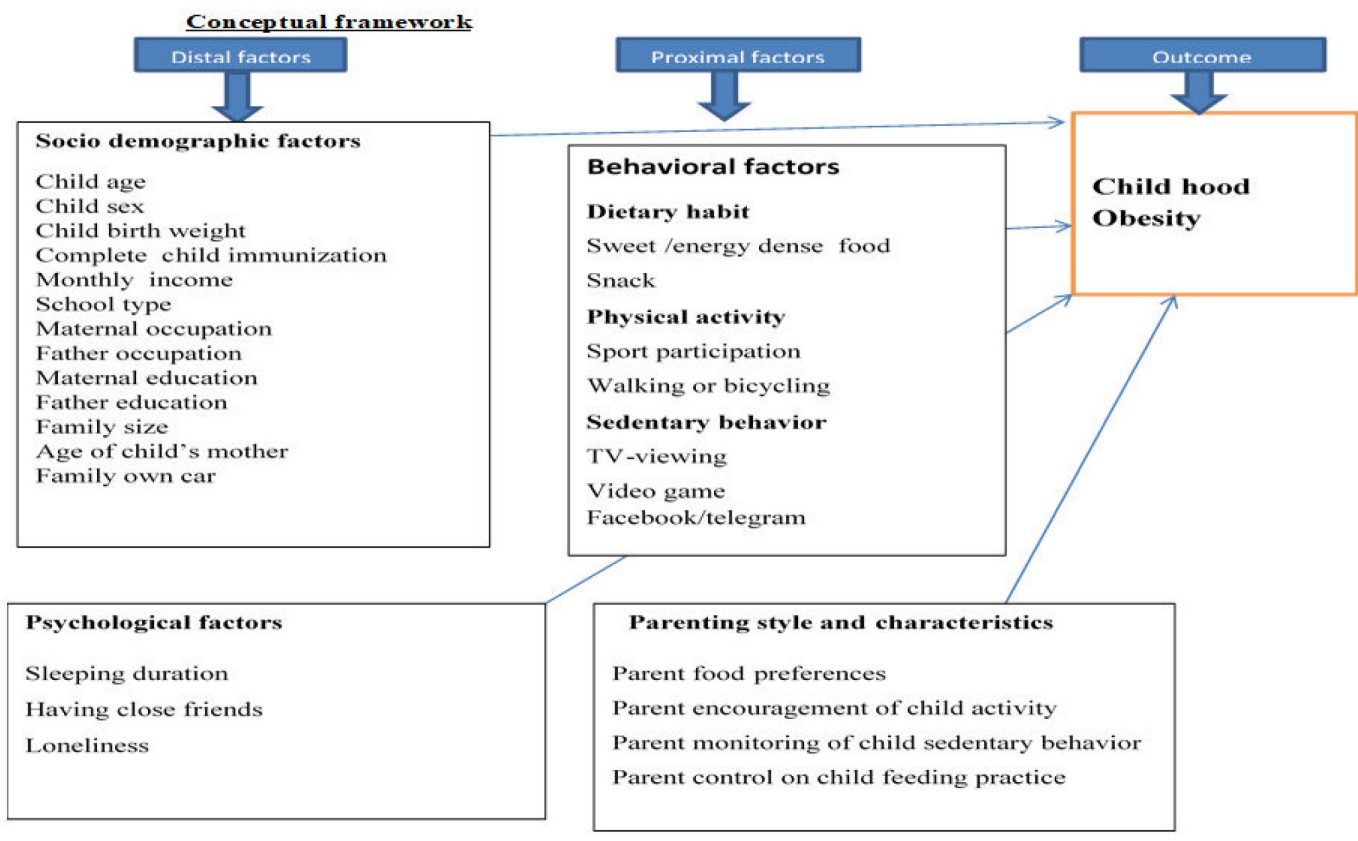

**Fig 1. Conceptual framework to assess the prevalence and associated factors of obesity among school age children in Debre Berhan City, Ethiopia, 2022 adapted from article(1–5).**

## Operational definitions

**Obesity.** BMI for age greater than or equal to 95th percentile (CDC, 2000).

**Dietary habit.** Refers to the number of days in which the children eat a particular food group (fruits, vegetables, meat, oil, sweet food, milk) in weeks in the past months at the time of data collection [2].

**Sedentary life.** Time spent on social media (Telegram, Facebook), and watching TV, or video playing or computer games for more than 2 hours per day [2].

**Total physical activity (TPA).** Total minutes spent on moderate to vigorous intensity physical activities per day at school or home, public physical activity facilities as a regular exercise, home works, recreation, and /or transportation [22].

- **Low TPA** = < 30 minutes TPA per day

- **Moderate TPA** = 30–59 minutes TPA per day

- **High TPA** = ≥ 60 minutes TPA per day

**Sleep duration.** Total sleep duration each day analyzed as ≥ 8hrs or <8 hrs of sleep based on national sleep foundation recommendation on sleep time.

**Loneliness.** total score computed by adding the response of UCLA scale with 10 item questionnaires, score 10–20 low loneliness, and score >21 high loneliness [23].

## Data collection procedures

Structured and pretested interviewer-administered questionnaires were used to collect data. The questionnaires included four sections: Socio-demographic characteristics of parents and children, behavioral factors, psychological related factors, and parenting style and family characteristics and height and weight measurement. Physical-activity related questions were adapted from the Global Physical Activity Questionnaire (GPAQ) analysis guide [24]. Dietary habit questions were adapted from the Food and Nutrition Technical Assistance [25]. The questionnaires were first developed the English version and translated in to the local Amharic language and reviewed by language experts for consistency of the language before data collection. The study participates were recruited from June 8 up to 12, 2022G.C. Both the children and their parents were involved in the interview. Concerning family characteristics, parents were interviewed. For questions that were related and specific to the children's characteristics, children were interviewed.

All the participating students were interviewed at school outside the classroom to keep responding freely and correctly. During weight and height measurements, jackets, sweaters, shoes, bags, and hair ornaments were avoided to minimize measurement errors. Height was measured to the nearest 0.1 cm in standing position at the Frankfurt plane with the occipital, shoulder and buttock touching the vertical stand using a stadiometer. The weight scale was calibrated at zero with no object on it and placed on the level surface before the measurement was carried out. Weight was measured by using a digital balance scale and recorded to the nearest 0.1 kg, and BMI-for-age Z-score (BAZ) was generated for each child using WHO Anthroplus version 1.0.4 software. Data were collected by four experienced BSc nurses and the data collector was supervised by one MSc nurse and principal investigator.

## Data quality control and assurance management

The questionnaire was prepared first in English and translated to the Amharic language and back to the English language. It was reviewed by language experts for consistency of the language and nutritionists to check its appropriateness for assessing overweight. Data collectors and supervisors were trained for two days about the whole procedures of data collection. The data were collected after 5% of the samples pretest was conducted. Then unclear questions were corrected and unnecessary questions were excluded based on the pretest. The investigators and supervisors had day-to-day supervision during the whole period of data collection. The weighting scale was calibrated and placed on the level surface before and after each measurement. Three measurements were taken for a single child and an average of three measurements were used for analysis. Continuous checkup of scales was carried out throughout the data collection. Valid instruments were used. The internal consistence of loneliness score questionnaire was determined by Cronbach's alpha test which was 0.98.

## Data analysis procedures

The collected data were first checked for completeness, clean and coded. Then, data were entered in to Epi-data version 4.6 and exported to SPSS version 25 software for cleaning and statistical analysis. Prevalence of overweight and obesity was determined by exporting the age, sex, height, weight of the child to WHO Anthro-plus version 1.0.4, and then BAZ were imported it to SPSS software for analysis. The dependent variable was recoded to dichotomous out comes as children with BAZ less than 85th percentile were coded as "0" and those greater than 85th percentile were coded as '1'. Independent variables were coded based on previous related studies and the distribution of responses to the data. Categorical variables were described using frequency, percentage, table, and figures. Bivariable logistic regression

analyses were used and Crude Odd Ratio (COR) with 95% CI was computed to assess the association between each predictor and the outcome variables. Variables with a p-value <0.25 during the bivariable analysis were included in the multi-variable logistic regression analysis. Multicollinearity between independent variables were checked using Variable Inflation Factor (VIF), and no significant (mean VIF = 1.65) colinearity was detected. Model goodness of fit was checked by Hosmer-Lemeshow test, and the final model was fitted (p-value = 0.75). Adjusting odds ratio (AOR) with 95% CI was estimated to identify the associated factors. Finally, statistical significance was declared at p value less than 0.05.

**Ethical consideration.**   Ethical approval and clearance were obtained from the institutional review board (IRB) of Asrat Woldeyes Health Science Campus. Letters of cooperation were given at all levels to the respective administrative officials. After obtaining permission from Debre Berhan city education office, written consent was obtained from the parents of the study participant, after informing them all the purpose, benefits, and voluntary natures of the participation in the study. Then verbal assent was obtained from the children. All information obtained from the study participants would be kept private and confidential. Codes and aggregate reporting were used to eliminate names and other personal identifiers of respondents throughout the study process to ensure anonymity.

## Result

### Socio-demographic characteristics of the parents

A total of 600 parents took part in the study, with a response rate of 98.04%. The mean (±SD) age of the child mothers was 36(±4.51) years. Some 378 (63%) of respondents were female and 501 (83.5%) attended education college and above. The mean (±SD) average family monthly income was 10482.1 (± 3575.5) Ethiopian birr. Three-hundred sixty-six (61%) of the mothers were government employees and 223(37.2%) of the fathers had private businesses. Ninety three (15.5%) of them had a family own car for transporting their child from school and 512 (85.3%) of them had less than or equal to five in their family size shown Table 1.

**Socio-demographic characteristics of the children.**   Among 600 children included in the study, two hundred sixty three (43.8%) and 337 (56.2%) of them were male and female respectively. The mean (±SD) ages of the children were 9.5 (±1.73) years old. The mean (± SD) of BAZ forage was 1.08(±1.66), weight 36.3((±8.40) kilograms (kg), and height 135.5(±8.14) centimeter respectively. Concerning the school type majority of (50.7%) and (49.3%) of the children were from government and private schools respectively seen in Table 2.

**BMI of study participants in comparison to WHO curve.**   BMI-Z Score distribution of children as compared to the WHO standard reference curve for 6–12 years at Debre Berhan city, Ethiopia, June to July, 2022 shown in **Fig 2**.

**Dietary habit and psychological factors.**   The dieting habit of participants shows that two hundred four (34%) and 206 (34.3%) of them did not consume fruits and vegetables respectively. Concerning their snack utilization, four hundred forty one (73.5%) of them used snacks and 262 (59.4%) two times and 176(40.5%) used three and more times per day. The majority of the respondents (96.2%) had their lunch by going to home. Regarding sleeping habits of children majority of them (58.5%) had the habit of daytime nap-taking see in Table 3.

**Physical activity and sedentary life style.**   Out of 600 participants, 434(73.2%) of the children did not participate in any work besides learning. Among children who participate in work, thirty two (5.3%) did Vigorous intensity work for at least 10 minutes. While 49 (8.2%) of them participated in moderately intense work for at least 10 minutes per day. Two hundred four (34%) had habits of walking or riding a bicycle. Concerning the sedentary behavior of the participants, two hundred forty one (40.2%), 225(37.5%) and 116(19.3%) spent their free time

**Table 1. Socio-demographic characteristics of parents among school- age children in Debre Berhan city, Ethiopian, June to July, 2022.** (n = 600).

| Variable | Category | frequency | Percentage |
|---|---|---|---|
| Respondent sex | Female | 378 | 63 |
| | Male | 222 | 37 |
| Age of child's mother | < 30 years | 78 | 13 |
| | 31–35 years | 186 | 31 |
| | 36–40 years | 273 | 45.5 |
| | .> = 41 years | 63 | 10.5 |
| Educational status of mother | Able to read &write | 18 | 3 |
| | Primary school | 18 | 3 |
| | Secondary school | 63 | 10.5 |
| | College and above | 501 | 83.5 |
| Educational status of father | Can't read and write | 10 | 1.7 |
| | Able to read &write | 14 | 2.3 |
| | Primary school | 25 | 4.2 |
| | Secondary school | 57 | 9.5 |
| | College and above | 494 | 82.3 |
| Occupation of the Mothers | House wife | 80 | 13.3 |
| | Government Employee | 366 | 61 |
| | Private Business | 135 | 22.5 |
| | daily laborer | 19 | 3.2 |
| Occupation of the Fathers | Government Employee | 324 | 54 |
| | Private Business | 223 | 37.2 |
| | daily laborer | 53 | 8.8 |
| Estimated monthly income | <5000 ETB | 29 | 4.8 |
| | 5001–10000 ETB | 227 | 37.9 |
| | > 10001 ETB | 344 | 57.3 |
| Family size | < = 5 | 512 | 85.3 |
| | >5 | 88 | 14.7 |
| Family own car | Yes | 93 | 15.5 |
| | No | 507 | 84.5 |

**Table 2. Socio-demographic, characteristics of children among school- age children in Debre Berhan city Ethiopian, June to July, 2022.** (n = 600).

| Variable | Category | Frequency | Percentage |
|---|---|---|---|
| Sex | Female | 337 | 43.8% |
| | Male | 263 | 56.2% |
| Age | 7–9 years | 188 | 31.3 |
| | 10–12 years | 412 | 68.7 |
| Grade | One | 86 | 14.3 |
| | Two | 91 | 15.2 |
| | Three | 99 | 16.5 |
| | Four | 92 | 15.3 |
| | Five | 118 | 19.7 |
| | Six | 114 | 19.0 |
| School type | Private | 296 | 49.3 |
| | Government | 304 | 50.7 |
| Complete Immunization status | Yes | 361 | 60.2 |
| | No | 239 | 39.8 |

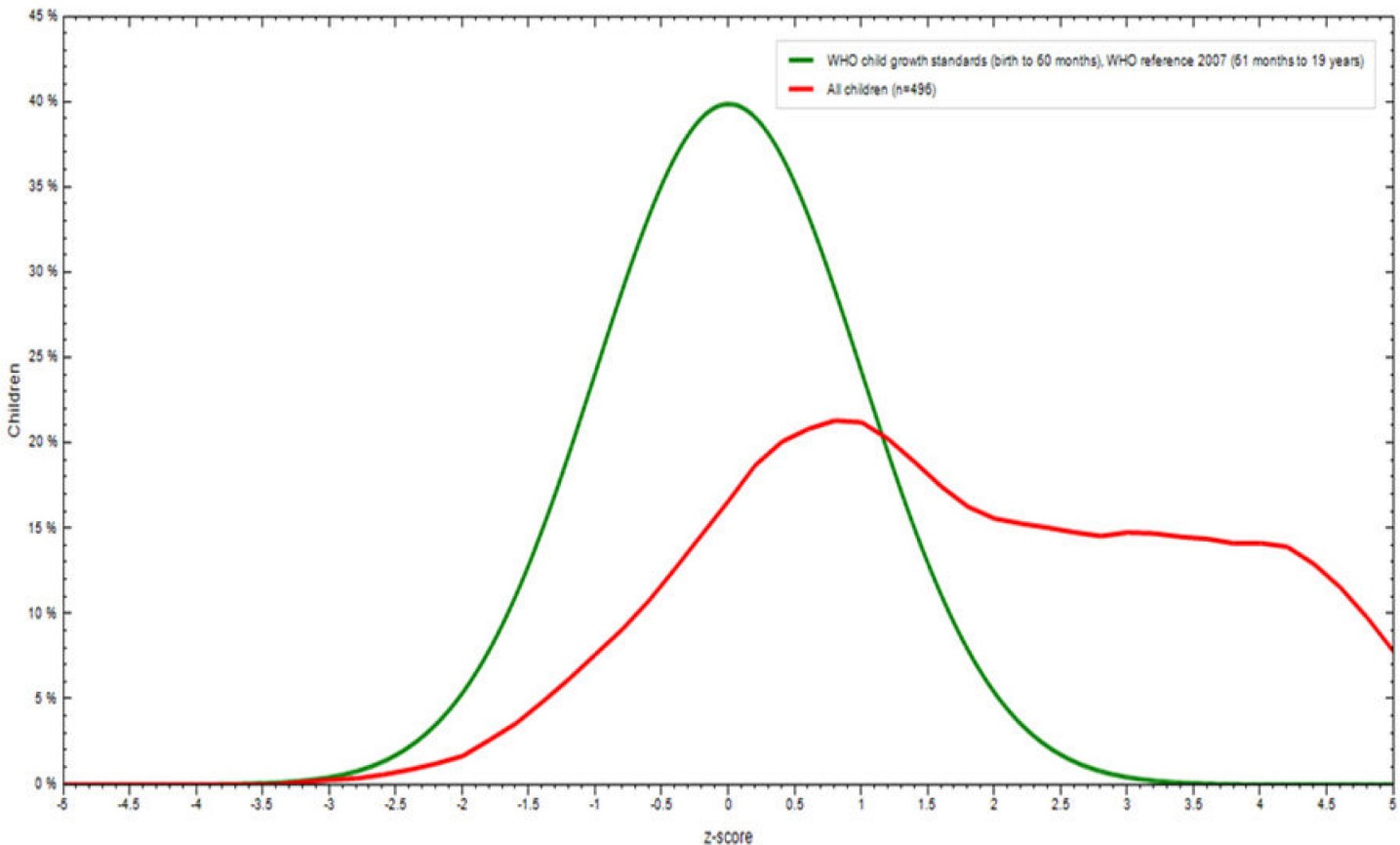

**Fig 2. BNII-Z Score distribution of children as compared to the WHO standard reference curve for 6–12 years at Debre Berhan city, Ethiopia.** June toJuly,2022 (n = 600).

sitting on face book or telegram, watching television, and playing computer video games respectively as shown in Table 4.

**Parenting style and family characteristics.**   One hundred seventy eight (29 .7%) of the participant eats at least one meal together every day and ninety eight (16.30%) of study subjects eat three to four days per week on the common plate with their parents. The majority (88.2%) of the parent-reported not having adequate space and 369 (61.5%) of the parents did not encourage their children to do physical activity see in Table 5.

**Magnitudes of obesity.**   The overall magnitude of underweight, normal weight, over- weight and obesity was 81 (13.5% with 95% CI; 10.8, 16.3), 288 (48.0% with 95% CI: 43.8, 52.0), 167(27.8% with 95% CI: 24.2, 31.5) and 64 (10.7% with 95% CI: 8.3, 13.2) respectively shown in **Fig 3**.

**Multivariate logistic regression.**   Data were analyzed using binary logistic regression analysis. Statistical associations were checked by 95% CI and odds ratio. Those variables which had a p-value less than 0.25 in the binary logistical regression analysis were eligible for multi- variable logistic regressions. Finally, the adjusted odds ratio was checked and the significant variables p value<0.05 were considered as associated factors for child hood overweight.

Based on the multivariable logistic regression analysis of this study, attending at private school, children aged between 10–12 years, soft drink available, loneliness, and mothers with occupational status of daily labour were significantly associated with childhood obesity.

**Table 3. Dietary habits& psychological factors among school-age children in Debre Berhan City, Ethiopia, June to July, 2022 (n = 600).**

| Variables | Category | Frequency | Percentage |
|---|---|---|---|
| have a snack | Yes | 441 | 73.5 |
| | No | 159 | 26.5 |
| Ways of getting lunch | Bring from home | 577 | 96.2 |
| | Buy from school cafeteria | 10 | 1.7 |
| | nearby food service | 13 | 2.1 |
| Eat While Watching television | Yes | 483 | 80.5 |
| | No | 117 | 19.5 |
| Eat breakfast irregularly | Yes | 256 | 42.7 |
| | No | 344 | 57.3 |
| Eat while studying | Yes | 400 | 66.7 |
| | No | 200 | 33.3 |
| fruit consumption per week | did not consume | 204 | 34.0 |
| | 1–2 days per week | 178 | 29.7 |
| | > = 3 days per week | 218 | 36.3 |
| Vegetable consumption per week | did not consume | 206 | 34.3 |
| | 1–2 days per week | 178 | 29.7 |
| | > = 3 days per week | 216 | 36.0 |
| Frequency of snack consumption per day | < = 2 times | 262 | 59.4 |
| | > 2 times | 179 | 40.6 |
| Number of meals other than snack | < = 3 times | 128 | 21.3 |
| | > = 4 times | 472 | 78.7 |
| Day time nap taking | <30 minutes | 73 | 12.2 |
| | 30-60minutes | 351 | 58.5 |
| | >60 minutes | 176 | 29.3 |
| Sleep duration per day | <8 hours | 119 | 19.8 |
| | 9–11 hours | 256 | 42.7 |
| | > = 12 hours | 225 | 37.5 |
| Loneliness | Low loneliness | 256 | 42.7 |
| | High loneliness | 344 | 53.3 |

Children aged between 10–12 years were 2.67 times more likely to be obese compared to those whose age between 7–9 years (AOR = 2.67, 95% CI: 1.30, 5.48). This finding suggests that mothers with an occupation status of daily labour were almost 8.5 times more likely to have an obese child compared to those housewives (AOR = 8.54 95% CI: 1.12, 65.39). Also, this finding indicates that soft drink available in home is 2.27 times more likely obese compare to do not have (AOR = 2.27, 95% CI: 1.25, 18.13).The finding states children with higher loneliness are 1.67 times more likely obese compare to low loneliness (AOR = 1.67 95% CI: 1.12, 3.15). Similarly children who attend in Private schools were 4.24 times more likely to be obese compared to who attend a government schools (AOR = 4.24, 95% CI: 1.58, 11.39) as described in Table 6.

## Discussion

In this study, the overall prevalence of obesity was (10.7%). Attending at private school, children aged between 10–12 years, soft drink available, loneliness, and mothers with occupational status of daily labour were significantly associated with childhood obesity.

This study is consistent with a study in India revealed that obesity 11.5% [26]. The finding of this study showed that the magnitude of obesity in males (11.4%) were more than females

**Table 4. Physical activity and sedentary behavior among school-age children in in Debre Berhan city, Ethiopia, June to July, 2022 (n = 600).**

| Variables | Category | Frequency | Percentage |
|---|---|---|---|
| Engaged in Work besides your education | Yes | 166 | 27.7 |
| | No | 434 | 72.3 |
| Vigorous intensity activity for at least 10 minutes continuously | Yes | 32 | 5.3 |
| | No | 568 | 94.7 |
| Moderate- intensity activity for at least 10 minutes continuously | Yes | 49 | 8.2 |
| | No | 551 | 91.8 |
| Walk or use a bicycle for at least 10 minutes continuously | Yes | 204 | 34.0 |
| | No | 396 | 66.0 |
| Spending free time | Face book/telegram | 241 | 40.2 |
| | Watching TV/ film | 225 | 37.5 |
| | Computer games | 116 | 19.3 |
| | Others | 18 | 3.0 |
| Time spent on face book | < = 120minutes | 541 | 90.2 |
| | >120 minutes | 59 | 9.8 |
| Time spent on video games | < = 120minutes | 537 | 89.5 |
| | >120 minutes | 63 | 10.5 |
| Time spent on watch TV | < = 120minutes | 481 | 80.2 |
| | >120 minutes | 119 | 19.8 |

**Table 5. Parenting style and family characteristics among school-age children in Debre Berhan city, Ethiopian June to July, 2022 (n = 600).**

| Variable | Category | Frequency | Percentage |
|---|---|---|---|
| family eat at least one meal together each day | never | 88 | 14.7 |
| | 1–2 times in a wee | 236 | 39.3 |
| | 3–4 times in a week | 98 | 16.3 |
| | Daily | 178 | 29.7 |
| family eat fruits and/or vegetables with your main meal | never | 40 | 6.7 |
| | 1–2 times in a week | 280 | 46.7 |
| | 3–4 times in a week | 134 | 22.3 |
| | Daily | 146 | 24.3 |
| family have adequate space for the children to play | Yes | 71 | 11.8 |
| | No | 529 | 88.2 |
| family encourage the child to be physically active or play sports | Yes | 231 | 38.5 |
| | No | 369 | 61.5 |
| family have any firm limits watch (TV, mobile) or video games | Yes | 257 | 42.8 |
| | No | 343 | 57.2 |
| Soft drinks available in your home | Yes | 275 | 45.8 |
| | No | 325 | 54.2 |
| Child snack (on chips, biscuits) or drink soft drink whenever they like | Yes | 270 | 45.0 |
| | No | 330 | 55.0 |
| Parents offer your child sweets to your child as | never | 140 | 23.33 |
| | some times | 29 | 4.83 |
| a reward for good behavior | usually | 431 | 71.83 |
| keep track of the high fat foods that your child eats | never | 383 | 63.8 |
| | some times | 78 | 13.0 |
| | usually | 139 | 23.2 |

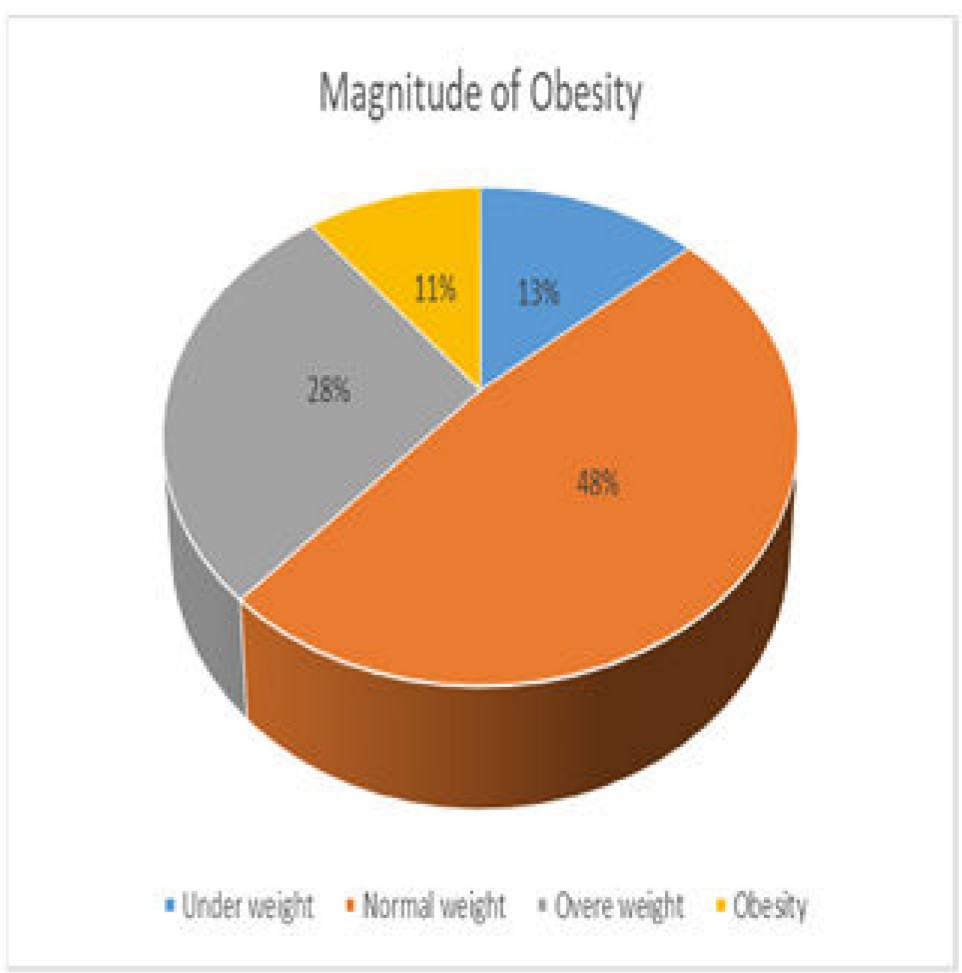

**Fig 3. Nlapitude of BNIE statues among, school age children Debte-berhan city.** Amhara r egi on Ethiopia 2022.

(10.1%) which was congruent with study in Addis Ababa reveled males were more obese (8.6%) than females (3.8%) [2]. However, in Nepal boys (19.0%) were overweight or obese than girls (10.6%) [1].These differences are most likely due to gender-based attitudes, as a female identity is typically defined by eating a smaller portion and preferring healthier options to maintain appearance, whereas a male eating identity is defined by feeling full while optimizing physical performance. In addition, females may prefer foods that are lower in energy and nutrient density, whereas boys tend to consume foods that are higher in energy and calorie density [27].

The finding of this study was lower in magnitude of obesity with a study done in Trinidad and Tobago revealing that 19.8% of obesity [28]. Furthermore, the findings of this study was also lower than those of previous studies conducted in Bangladesh which revealed 17.9% of obesity [6]. This may be explained by differences in life style, eating habits, socioeconomic status, and cultural factors. Hence, developed countries have different information options or media, children spend most of their time playing videos or computer games, and their nutritional habits are often packed with sweet and energy-dense foods.

The finding of this study was also lower in magnitude of obesity than the studies done in African countries like in Egypt, Kenya, and Tanzania, 14.6%, 25.6% and 11.4% respectively

**Table 6. Predictors for childhood obesity among school-age children in Debre Berhan city, Ethiopia, June to July, 2022.** (n = 600).

| Variables | Frequency | Obesity | COR(95% CI) | AOR(95%CI) | p-value |
|---|---|---|---|---|---|
| | | Yes No | | | |
| Mother    Housewife | 80 | 4 76 | 1 | 1 | |
| occupation    government | 366 | 39 327 | 2.27(0.78, 6.53) | 2.38(0.75, 7.57) | 0.142 |
| private business | 135 | 19 116 | 3.11(1.01, 9.50) | 2.15(0.65, 7.10) | |
| daily labor | 19 | 2 17 | 2.24(1.78,13.21) | 8.54(1.12,65.39)* | 0.21 |
| | | | | | 0.03 |
| Child age in years    6–9 | 188 | 12 176 | 1 | 1 | |
| 10–12 | 412 | 52 360 | 2.12(1.10, 4.07) | 2.67(1.30,5.48)* | 0.01 |
| School type    Private | 296 | 53 243 | 5.81(2.96, 11.36) | 4.24(1.58,11.32)* | 0.004 |
| Government | 304 | 11 293 | 1 | 1 | |
| Parent encourage the    Yes | 231 | 12 219 | 1 | 1 | |
| child physically active    No | 369 | 52 317 | 2.99(1.56, 5.74) | 2.83(0.67, 11.75) | 0.15 |
| Parent have any firm    Yes | 257 | 14 243 | 1 | 1 | |
| limit watch /TV,DVD,    No videogame) | 343 | 50 293 | 2.96(1.59, 5.48) | 0.65(0.148,2.84) | 0.566 |
| Soft drinks available    Yes | 275 | 42 233 | 2.48(1.44,4.27) | 2.27(1.25,18.13)* | 0.004 |
| in home    No | 325 | 22 303 | 1 | 1 | |
| Child snack (chips,    Yes | 270 | 41 239 | 2.39(1.39, 4.09) | 0.37(0.04, 3.45) | 0.387 |
| biscuits)    No or soft drink | 330 | 23 307 | 1 | 1 | |
| Have snack    Yes | 441 | 52 389 | 1.64(0.85, 3.15) | 0.58(0.22,1.51) | 0.586 |
| No | 159 | 12 147 | 1 | 1 | |
| Eat beak fast    Yes | 256 | 41 215 | 2.66(1.55, 4.56) | 1.15(0.56,2.19) | 0.75 |
| irregularly    No | 344 | 23 321 | 1 | 1 | |
| Moderate intensity    Yes | 49 | 2 47 | 1 | 1 | |
| activity    No | 551 | 62 489 | 2.89(0.71, 12.57) | 3.99(0.73,21.59) | 0.10 |
| Walk or use bicycle    Yes | 204 | 26 178 | 1.37(0.81, 2.34) | 0.80(0.43,1.51) | 0.499 |
| for at least 10 minutes    No | 396 | 38 358 | 1 | 1 | |
| Moderate intensity    Yes | 65 | 11 54 | 1 | 1 | |
| sports    No | 535 | 53 482 | 0.54(0.266,1.09) | 0.46(0.18,1.15) | 0.09 |
| Time spent <    2 hours | 541 | 55 486 | 1 | 1 | |
| on face book >    2hours | 59 | 9 50 | 1.59(0.74, 3.41) | 1.4(0.60,3.25) | 0.432 |
| Sleep duration    9–11 hrs | 256 | 36 220 | 1 | 1 | |
| per day    <8 hrs | 119 | 10 119 | 0.56(0.26,1.17) | 0.45(0.20,1.01) | 0.05 |
| > = 12 hrs | 225 | 18 207 | 0.53(0.29, 0.96) | 1.21(0.59,2.45) | 0.59 |
| Monthly income    <5000 | 29 | 6 23 | 1 | 1 | |
| In ETB    5001–10000 | 227 | 15 212 | 0.27(0.09,0.77) | 0.71(0.21,2.34) | 0.58 |
| > = 10001 | 344 | 43 301 | 0.55(0.21,1.42) | 0.97(0.34,2.79) | 0.96 |
| Numbers of meal    < = 3 times | 128 | 8 120 | 1 | 1 | |
| other than snack    > = 4 times | 472 | 56 416 | 2.02(0.94, 4.35) | 1.38(0.46,4.09) | 0.55 |
| Have close friends    Yes | 225 | 17 208 | 1 | 1 | |
| No | 375 | 47 328 | 1.75(0.98, 3.13) | 1.48(0.75,2.91) | 0.25 |
| Loneliness    Low | 256 | 17 239 | 1 | 1 | |
| High | 344 | 47 297 | 2.23(1.24,3.97) | 1.67(1.12, 3.15)* | 0.001 |

[22, 29, 30]. This might be explained by differences in feeding habits and socio-economic status as well as difference in standards for the cutoff point and sample size of participants. Furthermore, these countries are better economic status, the children may walk from home to school by family own car, the children frequently used sweet, and processed food, and also used different standards for the cutoff point and used smaller sample size of the participants. In this study, the magnitude of obesity were higher than studies done in Addis Ababa, which revealed that 7.5% obese [3]. This might be explained by differences lifestyle, age of the sample population and sample size. In addition, children in Addis Ababa better understand and appreciate the times spend in physical activity. The diet is often consists fruits and vegetables. It's also possible they are different ages and this study used a large sample size. This finding is also higher in magnitude of overweight and obesity than study in Bahir Dar (11.9%) of overweight and obesity [19]. This might be explained by differences due to study area, difference study period, and sample size.

Children aged between 10–12 years were significantly more likely to suffer from obesity than those children whose ages were between 7–9 years. This was supported by a study in Tanzania [31] which explained that age groups from10-12 years were more likely to have obesity. This could be due to the fact that the majority of children aged 10–12 years were at a pre-adolescent or adolescent stage. This is a stage when children attain a rapid growth spurt, characterized by rapid linear growth and the deposition of fat mass.

At this age, many children begin to be concerned about their own body images or shapes, and they frequently begin skipping meals and snacking on high-fat and high-sugar foods or drinks. Therefore, responsible bodies focus on creating awareness and providing extensive health education about the dietary habits of children, especially weight-reducing diets, protein-rich foods low in carbohydrate and fat, as well as consuming plenty of fruits and vegetables.

Mothers with an occupation status of daily labour were almost 8.5 times more likely to have an obese child compared to those housewives. This was similar to studies in the United Kingdom [32, 33], revealed that children on maternal employment supply were more likely to be obese. Several factors may explain how maternal employment is associated with overweight or obesity. For instance, working mothers could be pressed for time, which might cause them to substitute healthy, home-cooked meals for fast food and other complementary feeding. Maternal employment was associated with a short duration of breast feeding. This indicates that supplementary feeding at early age is a possible pathway through which children may gain weight. So that the stakeholders emphasis on programs encouraging healthy behaviors among children could be better tailored to bring both parents on board and to consider changes in family structures.

Children attending private schools were nearly 4 times more likely to be obese than those attending public schools. This was consistent with studies in different countries in Addis Ababa [2], Bahir Dar[19], Gonder [21], revealed learnt in private schools were more likely to be obesity. The possible explanation might be that the higher socioeconomic status of private school students would allow them a higher adoption of unhealthy nutritional habits (fast food, energy-dense snacks, and sweets) than other school students. Therefore, schools attention should be given to strengthening physical activities, providing nutritional education, and creating parents awareness about healthy diets, as well as other preventive measures.

Soft drink available in home was almost 2 times more likely obesity compare to who did not have. This was consistent with studies in Addis Ababa [2, 34]. The possible explanation might be that the as sweet food item is calorie dense food which results in positive energy balance to their consumers. Hence, reducing soft drink intake is a crucial preventive measure for childhood obesity.

Children with higher loneliness are 1.67 times more likely obese compare to low loneliness. This was in lined with studies in sub-Saharan Africa [35]. The plausible explanation might be disrupted sleep and alteration HPA-axis related to psycho-social stress may promote weight gain by increasing eating behaviour and abdominal obesity [36]. Therefore loneliness and social isolation of children should take attention.

## Conclusion

The overall magnitude of childhood obesity was 10.7% which is relatively high compare to the previous national study. Boys were more likely obese than girls. Children attending private schools were more obese than children attending government schools. Attending private school, children's aged between 10–12 years, soft drink available in home, loneliness, and mothers with occupational status of daily labour were the significant association with obesity. Therefore, if preventive measures are not implemented immediately, the prevalence of obesity among children may rise significantly, leading to chronic health problems.

### Recommendation

**For debre berhan health bureau.**

- More attention toward a healthy lifestyle, choosing a healthy food, and implementing preventive measures.

**For schools**

- Schools should provide nutrition education, instituting a required physical education environment to promote a healthy lifestyle as well as life skills lessons might be organized in the school.

**For researchers**

- Future researchers could use longitudinal studies to determine the cause-effect relationship between loneliness and childhood obesity as well as child age between 10–12 years and child hood obesity.

### Limitation of the study

There are some limitations of the thesis. Skin fold thickness measurement was not done, which might eliminate the limitation of BMI measurement. Another limitation, during interview, there might be social desirability bias by participants. There is no information about the relation of a father's age with childhood obesity.

## Supporting information

**S1 Data. SPSS data.**
(SAV)

**S1 File. English version questionary for obesity and its associated factors among school-age children in Debre Behan city, Ethiopia, 2022.**
(DOCX)

## Acknowledgments

We would like to thank the study participants, data collectors, and supervisors who were involved in this study and spent their valuable time responding to my study.

## Author Contributions

**Conceptualization:** Abebe Nigussie Ayele.

**Data curation:** Abebe Nigussie Ayele.

**Formal analysis:** Abebe Nigussie Ayele.

**Funding acquisition:** Abebe Nigussie Ayele, Abrham Demis Abayneh, Mitiku Tefera Haile.

**Investigation:** Abebe Nigussie Ayele, Abrham Demis Abayneh, Mitiku Tefera Haile.

**Methodology:** Abebe Nigussie Ayele, Abrham Demis Abayneh, Mitiku Tefera Haile.

**Project administration:** Abebe Nigussie Ayele, Abrham Demis Abayneh, Mitiku Tefera Haile.

**Resources:** Abebe Nigussie Ayele, Abrham Demis Abayneh, Mitiku Tefera Haile.

**Software:** AbdulWahhab Seid, Abrham Demis Abayneh, Mitiku Tefera Haile.

**Supervision:** Alemayehu Gonie Mekonen, AbdulWahhab Seid, Esubalew Guday Mitikie, Abrham Demis Abayneh, Mitiku Tefera Haile.

**Validation:** Alemayehu Gonie Mekonen, AbdulWahhab Seid, Esubalew Guday Mitikie, Abrham Demis Abayneh, Mitiku Tefera Haile.

**Visualization:** Alemayehu Gonie Mekonen, AbdulWahhab Seid, Abrham Demis Abayneh, Mitiku Tefera Haile.

**Writing – original draft:** Abebe Nigussie Ayele, Abrham Demis Abayneh, Mitiku Tefera Haile.

**Writing – review & editing:** Abebe Nigussie Ayele, Abrham Demis Abayneh, Mitiku Tefera Haile.

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
